# Biological Barriers for Drug Delivery and Development of Innovative Therapeutic Approaches in HIV, Pancreatic Cancer, and Hemophilia A/B

**DOI:** 10.3390/pharmaceutics16091207

**Published:** 2024-09-13

**Authors:** Emre Basar, Henry Mead, Bennett Shum, Ingrid Rauter, Cihan Ay, Adriane Skaletz-Rorowski, Norbert H. Brockmeyer

**Affiliations:** 1WIR—Walk In Ruhr, Center for Sexual Health & Medicine, Department of Dermatology, Venerology and Allergology, Ruhr-University Bochum, 44787 Bochum, Germany; adriane.skaletz-rorowski@klinikum-bochum.de; 2BioMarin LLC, San Rafael, CA 94901, USA; 3GenePath LLC, Sydney, NSW 2067, Australia; 4EMBL Australia Node in Single Molecule Science, School of Medical Sciences, University of NSW, Sydney, NSW 2052, Australia; 5Vaccentis AG, 8001 Zurich, Switzerland; 6Division of Haematology and Haemostaseology, Department of Medicine I, Medical University of Vienna, 1090 Vienna, Austria

**Keywords:** HIV, pancreatic cancer, hemophilia A and B, biological barrier, RNA interference, HIV microbicide, tumor microenvironment, AAV vector, valoctocogene roxaparvovec, etranacogene dezaparvovec

## Abstract

Biological barriers remain a major obstacle for the development of innovative therapeutics. Depending on a disease’s pathophysiology, the involved tissues, cell populations, and cellular components, drugs often have to overcome several biological barriers to reach their target cells and become effective in a specific cellular compartment. Human biological barriers are incredibly diverse and include multiple layers of protection and obstruction. Importantly, biological barriers are not only found at the organ/tissue level, but also include cellular structures such as the outer plasma membrane, the endolysosomal machinery, and the nuclear envelope. Nowadays, clinicians have access to a broad arsenal of therapeutics ranging from chemically synthesized small molecules, biologicals including recombinant proteins (such as monoclonal antibodies and hormones), nucleic-acid-based therapeutics, and antibody-drug conjugates (ADCs), to modern viral-vector-mediated gene therapy. In the past decade, the therapeutic landscape has been changing rapidly, giving rise to a multitude of innovative therapy approaches. In 2018, the FDA approval of patisiran paved the way for small interfering RNAs (siRNAs) to become a novel class of nucleic-acid-based therapeutics, which—upon effective drug delivery to their target cells—allow to elegantly regulate the post-transcriptional gene expression. The recent approvals of valoctocogene roxaparvovec and etranacogene dezaparvovec for the treatment of hemophilia A and B, respectively, mark the breakthrough of viral-vector-based gene therapy as a new tool to cure disease. A multitude of highly innovative medicines and drug delivery methods including mRNA-based cancer vaccines and exosome-targeted therapy is on the verge of entering the market and changing the treatment landscape for a broad range of conditions. In this review, we provide insights into three different disease entities, which are clinically, scientifically, and socioeconomically impactful and have given rise to many technological advancements: acquired immunodeficiency syndrome (AIDS) as a predominant infectious disease, pancreatic carcinoma as one of the most lethal solid cancers, and hemophilia A/B as a hereditary genetic disorder. Our primary objective is to highlight the overarching principles of biological barriers that can be identified across different disease areas. Our second goal is to showcase which therapeutic approaches designed to cross disease-specific biological barriers have been promising in effectively treating disease. In this context, we will exemplify how the right selection of the drug category and delivery vehicle, mode of administration, and therapeutic target(s) can help overcome various biological barriers to prevent, treat, and cure disease.

## 1. Introduction

Biological barriers such as the skin, the mucosa of the cervicovaginal tract and intestines, the blood–brain barrier, and the desmoplastic stromal tissue surrounding certain solid tumors remain major obstacles for the efficient delivery of innovative medicines.

The broad range of biological barriers can be divided into obstacles at a cellular/intracellular/molecular level and a (supracellular) tissue/organ level.

On a cellular level, the nuclear membrane, the lining of the organelles, various endosomal compartments, and the negatively charged outer lipid bilayer can be very difficult for drugs to cross.

Human organs and tissues pose additional challenges to the design and selection of an appropriate drug: in fact, poor access to the blood supply, the blood–brain barrier and blood–bone barrier, enzymatic degradation in the blood stream or inside the cellular endolysosomal machinery, and protection by the mucosal/epithelial lining can be major obstacles for therapeutics to reach their destination. 

Moreover, the genetic heterogeneity of pathogens and tumors, interception by the innate or adaptive immune system, and various immune evasion mechanisms can also form challenges for efficient therapeutic targeting.

This comprehensive review’s two objectives are to identify the overarching principles of biological barriers across different disease entities and to exemplify how the development of innovative drug delivery technologies helps overcome these barriers. 

For this purpose, we selected three different disease entities according to their global clinical, scientific, epidemiological, and socioeconomic relevance. Moreover, we chose disease areas which are suitable for showcasing a broad range of biological barriers and, at the same time, offer a multitude of technological novelties and breakthroughs. Based on these criteria, we decided to divide this review into disease-specific chapters that offer deep insights into the following three disease entities: Chapter I focuses on the biology of human immunodeficiency virus (HIV) as a one of the world’s predominant infectious diseases, and on novel therapeutic strategies to fight acquired immunodeficiency syndrome (AIDS). Chapter II visits pancreatic cancer as one of the most lethal solid tumors whose incidence is rising in many countries. Here, we highlight the biological hallmarks of pancreatic cancer, all of which represent challenging biological barriers, and offer a glimpse into the huge range of therapeutic approaches. In Chapter III, we introduce hemophilia A/B as a rare monogenetic disease and discuss how recent advancements in gene therapy have revolutionized the treatment in this field.

In all three chapters, we describe the biological barriers which have hampered the development of effective therapies. At the same time, we provide insights into new technological advancements that enable us to overcome these barriers to prevent, treat, or cure disease.

## 2. Chapter I: HIV/AIDS—Biological Barriers in HIV and Innovative Therapeutic Strategies to Prevent and Treat HIV Infection

### 2.1. Introduction to HIV/AIDS

After the emergence of HIV/AIDS in 1981, the spread of HIV rapidly evolved into the world’s most prevalent epidemic, causing significant morbidity and mortality [1,2]. Today, about 37–39 million people worldwide carry the HIV virus, and about 1.3 million people are estimated to be infected with HIV every year, with heterosexual transmission accounting for the majority of new infections [3,4,5,6].

HIV is a lymphotropic retrovirus which belongs to the genus of lentiviruses [7]. It can be classified into two strains, HIV-1 and HIV-2, both of which originate from the simian immunodeficiency virus (SIV) circulating in non-human primates [8]. Phylogenetically, HIV-1 can be further classified into four lineages (M, N, O, and P), with each group being the result of an independent cross-species transmission in central western Africa [5,9]. Representing 90–95% of all HIV-1 infections worldwide, the highly infectious HIV-1 M virus is the predominant strain and main causative agent of the AIDS pandemic [10,11].

Structurally, HIV can be divided into a lipid membrane (envelope) as its outer layer, an underlying matrix, and a cone-shaped capsid at its core [5]. The HIV genome consists of two identical single (positive-sense) RNA strands which are encased by the capsid. The outer layer of the virus is made up of a phospholipid bilayer membrane: it harbors two surface glycoproteins, gp120 and gp41, which are critical for the interaction with the host target cell receptors [12].

HIV-1 primarily targets human CD4^+^ immune cells which simultaneously express one of the two co-receptors (CCR5 or CXCR4) that are required for cell entry. Therefore, T-helper lymphocytes, macrophages, and dendritic cells (DCs), including Langerhans cells, are the three major targets of HIV-1 [13]. 

Upon infecting the host’s immune cells, HIV-1 takes advantage of a unique set of three replication enzymes that are vital for sustaining the HIV life cycle: reverse transcriptase (RT), integrase (IN), and protease (PR) [14,15]. Upon target cell binding and intracellular uptake, HIV exploits the invaded cells’ transport mechanisms to reach the nucleus [7]. By harnessing the enzymatic activities of RT and IN, HIV hijacks the host cell’s replication machinery and protein synthesis capabilities, which results in highly efficient viral proliferation and reproduction. In the final step of its life cycle, PR orchestrates the assembly of viral particles which are finally released from their host cells to infect new target cells.

In terms of disease manifestations, the (productive) infection of CD4^+^ T-helper cells is the most crucial event, as it results in T-cell depletion, which manifests in a continuous decline in the immune system’s functionality and culminates in its collapse. Ultimately, by weakening the host’s immune system by CD4^+^ T-cell depletion, HIV deprives the human body of a potent weapon to fight infection [16,17,18].

The success of HIV in advancing to a worldwide pandemic and remaining one of the globally most significant socioeconomic burdens for more than four decades is based on several hallmarks:

As a retrovirus, HIV has the ability to integrate into the host cell’s DNA to form a provirus [7]. It then replicates by exploiting the host transcriptional and translational machinery.

An important hallmark of HIV is its genetic heterogeneity, reflected by the existence of numerous HIV strains spread all over the world. The genetic diversity of HIV-1 is rooted in the unusually high error rate during the reverse transcription of viral RNA, the enormous replication activity of viral reverse transcriptase, the extensive recombination upon transcription from (viral) DNA, and the vast number of virions generated in each patient [19,20]. 

From an immunological point of view, the characteristics of HIV are its profound infiltration of the immune system and the knockout of key components of the host’s immune surveillance, which results in immune evasion [13]. Moreover, HIV causes chronic inflammation which—partly attributed to its latency in persisting reservoirs—further weakens the immune system [21,22].

All these hallmarks form diverse biological barriers (summarized in Figure 1), which—despite four decades of intense research—have been challenging to overcome. Although the development of an effective HIV vaccine to eradicate the pandemic remains elusive, several therapeutic approaches have evolved which dramatically reduced the impact of HIV on morbidity and mortality.

### 2.2. ART/HAART

The most important concept that revolutionized HIV treatment is antiretroviral therapy (ART): in fact, the emergence of systemically applied ART has transformed HIV from a deadly disease into a chronic condition [2,23,24].

Insights into the viral mechanisms of integration and subsequent replication allowed the development of antiretroviral therapeutics which attack key points of the HIV life cycle. By inhibiting the viral enzymes reverse transcriptase, integrase, and protease, antiretroviral drugs paralyze HIV’s replication machinery and hence dramatically reduce the viral burden [2].

However, ART is challenged by the fast transformation of HIV, which is a major factor for drug resistance: as HIV’s error-prone reverse transcriptase has no proofreading activity, it gives rise to escape mutations which often lead to treatment resistance to a single antiretroviral drug [25]. Therefore, patients are treated with a combination antiretroviral therapy (cART), which includes at least three drugs directed at two or more different targets within the HIV life cycle [2].

This combination therapy, called highly active antiretroviral therapy (HAART), has evolved into the gold standard in the treatment of HIV-infected individuals. In fact, HAART reduces the viral load to almost undetectable levels, dramatically decreases the risk of viral transmission during sexual intercourse, extends the lifespan by reducing mortality, and increases the quality of life by inhibiting the emergence of AIDS including its opportunistic infections [2,24,26].

For the treatment of HIV-infected patients, HAART has several limitations: on one hand, it requires lifelong daily oral intake of medication, which is why compliance remains a key challenge. On the other hand, it does not offer a cure for HIV infection, because HIV virions persist in latent reservoirs [24,26].

Being highly effective, combination ART is not only indicated for patients infected with HIV, but it is also used as pre-exposure prophylaxis (PrEP) or post-exposure prophylaxis (PEP) to prevent HIV infection prior to or after sexual intercourse with an HIV-infected individual [27,28,29,30,31].

The main drawbacks of PrEP and PEP as preventive tools are the lack of compliance and their challenging affordability in developing countries [28].

### 2.3. HIV Vaccine

Despite the great success of modern combination antiretroviral therapy, namely HAART for the treatment of HIV patients and PrEP/PEP for the prevention of HIV transmission upon sexual intercourse, the development of an effective and safe vaccine remains elusive [16,32]. In fact, the research community is convinced that only a preventive globally available HIV vaccine can eradicate the pandemic [16].

Several biological barriers impede the development of an HIV vaccine: first, the high mutation rate during reverse transcription results in an extraordinary diversity of HIV strains and thus transforms HIV into a continuously changing immunological target [16].

Second, HIV has the ability to evade adaptive immune responses, which is partly attributed to the fact that it infiltrates and invalidates the immune system. Moreover, HIV establishes latent viral reservoirs in various immune cells which are difficult to target [16]. Recently, vaginal epithelial cells have been shown to contribute to viral sequestration and subsequent dissemination, hence adding a new layer of complexity into treating viral latency [6,13].

Finally, it turned out to be challenging to develop antibodies which do not only recognize highly immunogenic (antigenic) epitopes but also induce a broadly reactive antibody response. The HIV envelope glycoprotein complex, which—formed by the surface glycoprotein gp120 and the transmembrane fragment gp41—is embedded in the viral envelope, would be a formidable druggable target. However, gp120 is heavily glycosylated, which is why large parts of the antigenic epitopes of the glycoprotein complex are shielded by sugar moieties. Epitope masking due to the heavy glycosylation of the HIV envelope glycoprotein forms a powerful biological barrier and is one of the main reasons why the development of effective antibodies has not been successful so far [16,33].

Novel vaccine models focus on either designing immunogens, which induce broadly neutralizing antibodies (bnABs) to bind conserved epitopes on HIV envelope glycoproteins, or harnessing cytomegalovirus vectors to elicit a cellular CD8^+^ T-cell-mediated immune response [17,34].

### 2.4. HIV Microbicide

In the absence of a preventive effective vaccine, new therapeutic approaches are required to limit the spread of HIV. As heterosexual transmission accounts for most new infections, the development of a preventive HIV microbicide to inhibit vaginal HIV transmission is considered to be a promising avenue [35].

The structure and dynamics of the female genital tract (FGT) play a pivotal role for infectivity with HIV and other sexually transmitted infections [36,37]. The female reproductive tract (FRT) can be divided into a lower part including the vaginal mucosa and ectocervix, and an upper part, which comprises the endocervix, uterus, fallopian tubes, and ovaries [38]. As the lower female tract is the primary site of contact with infectious seminal fluids, it offers multiple layers of protection: the apical layer of the vaginal epithelium is shielded by a protective lubricating mucus [36,38]. The mucosal epithelial lining is made up of a stratified squamous epithelium which is robust against mechanically induced lesions. Multiple intercellular connections such as tight junctions and desmosomes help decrease tissue permeability. The basal layer of the epithelium borders the underlying lamina propria (LP), which includes most HIV-susceptible immune cells such as T-cells, macrophages, and dendritic cells [36].

To reach their CD4^+^ target cells in the lamina propria and subsequently establish infection in the cervicovaginal tract (CVT), HIV-1 virions in the male semen must first overcome two important obstacles: the lubricant mucus covering the mucosa, as well as the underlying epithelial cell barrier. 

The mucus acts as a very first line of defense, as it can trap and destroy HIV virions. It contains many soluble factors such as immunoglobulins which recognize, bind, and eliminate invading pathogens [36,37,38]. Additionally, the low pH in the cervicovaginal cavity, which is induced by lactic acid production by lactobacilli, contributes to killing microbes and viral particles. In this context, vaginal epithelial cells (ECs) play a vital role in regulating the microbial hemostasis and maintaining the acidic milieu, as they produce and store glycogen which serves as a substrate for fermentation by lactobacilli [39,40,41].

Once virions in the (male) semen traverse the mucus as the first site of contact with the human body, they face the vaginal epithelium, which is challenging to cross. In fact, the stratified non-keratinized squamous epithelium constitutes a powerful physical barrier, as pathogens have to overcome several layers of epithelial cells. Moreover, epithelial layers are closely connected via tight junctions, which seal the space between cells to prevent intercellular pathogen passage [42].

Despite these layers of protection, HIV harnesses multiple mechanisms and routes to cross the epithelial layer. On the one hand, virions passage through the intercellular space until reaching HIV-susceptible immune cells in deeper layers of the epithelium and the underlying LP [43]. On the other hand, virions cross epithelial cells via transcytosis upon endosomal uptake at the apical cell membrane [13,43,44,45].

Importantly, it is not only CD4^+^ human immune cells that are targeted and infected by HIV. As emerging new evidence suggests, vaginal epithelial cells are also infected by HIV and actively contribute to viral dissemination throughout the FGT. Several studies reveal that, upon uptake, HIV virions are sequestered in the endosomal compartment (multivesicular bodies) of epithelial cells [6,46]. Afterwards, sequestered virions are passed on from ECs to susceptible immune cells via cell-to-cell transmission [47,48].

These groundbreaking insights into the mucosal pathogenesis of HIV infection open the door for exploring many new druggable targets in vaginal epithelial cells including genes mediating HIV binding (complement receptor 3, and heparan sulfate), endosomal uptake (clathrin, caveolin, GalCer, and TIM-1), transcytosis, sequestration, epithelial cell polarization, and cell-to-cell transmission to susceptible immune cells (ICAM-1) [13,43,44,45,47,48].

Over the past decade, studying the role of immune cells in vaginal HIV infection elucidated new routes of HIV transmission and viral dissemination. Druggable targets in immune cells go far beyond the established targets such as the CCR5/CXCR4 co-receptors that mediate HIV uptake into macrophages, dendric cells, and T lymphocytes. In fact, druggable targets can be expanded to genes/proteins involved in HIV attachment/binding (C- and i-type lectin receptors on DCs), endosomal uptake, the sequestration of virions in multivesicular bodies, the cell-to-cell transmission between immune cells (including factors necessary to establish filopodia in DCs; genes mediating actin nucleation; and proteins required to form infectious and viral synapses), the maturation and migration of immune cells (CCR7), and the release of cytokines which mediate cellular cross-talk and immune cell recruitment [49,50,51,52,53,54].

The same way the cervicovaginal mucosa acts as a biological barrier which HIV must penetrate to obtain access to its CD4^+^ target cells, the mucosa also impedes drug delivery to HIV target cells. Therefore, therapeutic strategies including the development of a vaginal HIV microbicide need to be designed in such a way that they can effectively overcome the vaginal mucosa as an important biological barrier.

Harnessing RNA interference (RNAi) technology might be a promising tool with which to target vaginal immune cells.

In 1998, Craig Mello and Andrew Fire described a fascinating phenomenon which they had discovered in the nematode worm *C. elegans* [55,56]. The following years led to the unraveling and deeper understanding of this evolutionarily conserved mechanism, which—termed RNA interference—allows to elegantly control gene regulation.

In 2001, Thomas Tuschl’s group showed that RNAi-mediated gene silencing was not only applicable in animals and plants, but also in mammalian cells. By using small interfering RNA duplexes (siRNAs) as RNAi effector molecules, they induced sequence-specific post-transcriptional gene silencing in mammalian cells and, hence, demonstrated the potential of siRNAs as a gene-specific drug for humans [57].

In contrast to gene therapy, which manipulates the cellular genome and, therefore, bears the risk of causing cancer, RNAi-based gene regulation happens on a mRNA level without interfering with the cellular DNA. Due to its elegant way of regulating gene expression, RNAi soon opened the door to a multitude of applications, both as a powerful research tool to study gene function and as a promising new class of nucleic-acid-based therapeutics [58].

Despite the great enthusiasm for harnessing RNAi as a therapeutic, the journey of developing RNAi-based drugs has been challenged by several obstacles. A major hurdle is to ensure that siRNAs are efficiently delivered to their target organs and cells. On a cellular basis, the major difficulties for siRNAs to become effective were to cross the negatively charged outer cell membrane, to escape the endolysosomal compartment, and to finally translocate to the cytoplasmic p-bodies as their site of action [59,60].

On a macroscopic level, the essential prerequisites for efficient siRNA delivery are a favorable pharmacokinetic profile, protection from degradation during transport through the bloodstream or across mucosal barriers, and the precise targeted delivery to the desired cell populations and diseased organs [61].

A series of advancements in siRNA chemistry and design contributed to resolving these challenges. Due to their phosphate backbone, siRNAs are negatively charged, which makes it difficult for them to cross the hydrophobic plasma membrane [62]. An early approach which turned out to be successful is to either embed naked siRNAs into lipid-based nanocarriers (liposomes, and lipoplexes) or conjugate them to cholesterol: this modification allows siRNAs to overcome the target cells’ plasma membrane, and subsequently—upon endosomal uptake—escape from the endosomal compartment to reach the p-bodies [59]. The incorporation of phosphorothioate residues into the siRNA design permits to stabilize them against nuclease degradation in the acidic environment of the vaginal mucosa [35,62].

Several research studies established a proof-of-concept that demonstrates the potential of a vaginally applied siRNA-based microbicide to inhibit heterosexual HSV-2 transmission [63,64,65].

Palliser et al. found that intravaginally (IVAG) instilled lipoplexed siRNAs targeting two viral genes protected mice from a lethal infection with herpes simplex virus 2 (HSV-2) [63]. In a subsequent study, Yichao Wu et al. showed that intravaginally applied cholesterol-conjugated siRNAs (chol-siRNAs) were efficiently taken up into the vaginal epithelium in mice. The IVAG administration of chol-siRNAs, which target HSV-2 viral genes as well as EC host genes required for viral entry, successfully inhibited HSV-2 infection in mice [64].

We wondered whether these results could be expanded to HIV. In comparison with HSV-2, the development of an RNAi-based vaginal microbicide against HIV is more challenging, as it requires drug penetration throughout the entire cervicovaginal mucosa to obtain therapeutic access to the HIV-relevant immune cell populations in the epithelial lining and underlying lamina propria. Although the siRNA-mediated transfection of immune cells proved to be difficult in vitro as well as in vivo, it is important to explore which siRNA formulations are suitable for inhibiting HIV uptake, viral sequestration, cell-to-cell transfer, and subsequent systemic dissemination.

Based on Yichao Wu’s promising findings, we embarked on a study that examined to which extent chol-siRNAs could be harnessed for gene silencing in HIV-relevant immune cells in the CVT [35,66]. Harnessing intravaginally instilled chol-siRNAs targeting the CD45 gene, we demonstrated (partial) gene silencing in mucosal dendritic cells including intraepithelial and lamina propria DCs [35].

Recent insights into the broad range of mechanisms which HIV exploits to cross the mucosal barrier to infect both vaginal epithelial and immune cells deliver a multitude of new gene targets that can be integrated into an RNAi-based microbicide.

Given the success of the above-mentioned proof-of-principle studies in mice, a mixture of siRNA formulations including chol-siRNAs could be harnessed to simultaneously block multiple routes of HIV transmission both in vaginal immune cells and ECs. Therefore, a promising approach would be to develop an RNAi-based microbicide based on multiplexes of chol-siRNAs that target a variety of host genes as well as conserved regions in viral genes to reduce or even inhibit the transmission of HIV in the FGT.

In conclusion, several treatment approaches have succeeded in limiting the HIV pandemic and transforming HIV into a chronic condition. Although several hallmarks of HIV, summarized in Figure 1, still form powerful biological barriers that impede more effective therapy, new preventive vaccine candidates and microbicides might soon become available to eradicate the pandemic.

## 3. Chapter II: PANCREATIC CANCER—Biological Barriers and Novel Innovative Therapeutic Approaches to Treat Pancreatic Cancer

### 3.1. Introduction to Pancreatic Cancer

Despite many significant therapeutic advancements including the emergence of innovative immune-therapy, cancer remains one of the leading causes of morbidity and mortality worldwide [67,68,69]. 

It is estimated that, in 2022, cancer gave rise to almost 20 million new cases and killed about 10 million people, turning it into the second biggest cause of death [68].

As a result of the aging population and increased exposure to risk factors, the global burden of cancer is anticipated to grow rapidly and reach about 29.9 million new cancer cases per annum by 2040, which translates into an increase of almost 50% from 2020 [70,71].

In the broad and heterogeneous spectrum of malignancies, pancreatic cancer (PC) stands out as one of the most threatening tumors with a dismal five-year survival rate of 8–12% [67,72,73,74]. Because of its high lethality, its current incidence translates into an almost equal mortality rate [72,75]. The American Cancer Society anticipates PC to surpass breast, colorectal, and prostate cancer to become the second leading cause of cancer-related death in the US, which reflects similar trends in all regions worldwide [70,73,76,77,78,79,80].

One important reason for pancreatic cancer to be such a deadly disease is attributed to its location. Deeply embedded into the abdominal retroperitoneal space and surrounded by many important organs and vessels, a surgical intervention is challenging—in particular, if the tumor has infiltrated neighboring organs. Because of its central location in the abdomen, many of the unspecific symptoms arising from PC can potentially be attributed to other abdominal organs so that the malignancy is often recognized and diagnosed in advanced (local or metastatic) disease stages [70,72,73,81]. As a result of its late detection, PC often turns out to have invaded its surrounding organs and/or spread to distant organs, thus rendering it inoperable [82,83].

A second reason for PC to be highly treatment resistant and lethal is based on its capability to form multiple biological barriers, which a drug needs to overcome in order to exert its therapeutic effects. In general, these barriers are the result of three characteristic hallmarks of PC, which contribute to its malignancy: a favorable tumor microenvironment (TME) which establishes a tumor-promoting immunosuppressive milieu, the tumor’s remarkable genetic heterogeneity, and its tendency for early advanced metastatic disease (summarized in Figure 2) [72,78,80].

Comprising about 90% of all pancreatic cancer cases, the histologically most relevant type is pancreatic ductal adenocarcinoma (PDAC), which originates from the exocrine cells that line the pancreatic duct [70,78,81]. Mutagenesis transforms normal pancreatic ductal cells into a preinvasive precursor lesion, a so-called pancreatic intraepithelial neoplasia (PanIN), which marks the first step in the development of PDAC [78,79,80]. Fostered by its genetic heterogeneity and nourishing tumor microenvironment, PDAC grows aggressively and quickly infiltrates neighboring as well as distant organs.

Despite many years of research and the development of a broad range of therapy modalities comprising surgery, radiation, and drug-based treatments (including standard chemotherapy, immunotherapy, and targeted therapy), PC remains resistant to current treatment approaches [72,81].

In the subsequent paragraphs, we will first provide more profound insights into the three above mentioned hallmarks of PC and then describe the arsenal of therapeutics which are applied to fight this deadly disease.

### 3.2. Tumor Microenvironment (TME) as a Biological Barrier

Pancreatic carcinoma has the ability to remodel the cellular, immune, vascular, and connective tissue components in its neighborhood, hence creating a favorable TME which promotes its development [69,79].

The hallmarks of pancreatic cancer’s complex TME comprise its cellular heterogeneity, continuous intercellular cross-talk, a desmoplastic reaction embedding the tumor into a fibrotic capsule, an abnormal vascular network resulting in poor blood supply, and an immunological pro-tumorigenic milieu (see Figure 2). In fact, the protective and nourishing tumor microenvironment of PC provides the foundation for many characteristics of its malignancy. At the same time, many challenges for a successful treatment can be attributed to the TME’s capacity to form a physical and immunologic barrier [69,79].

The dynamic ecosystem of the TME comprises multiple cell populations. Primarily, it includes a heterogeneous population of cancer cells (including cancer stem cells as well as senescent cells), which exhibit an enormous genetic and phenotypic diversity as drivers for tumor growth and treatment resistance [79]. Furthermore, the TME comprises a broad spectrum of immune cells such as tumor-associated macrophages (TAMs), T cells (CD4, TH17, and Tregs), B cells, NK cells, neutrophils, and myeloid-derived suppressor cells [76,79,81]. A third characteristic cell population of the TME are cancer-associated fibroblasts (CAFs) and pancreatic stellate cells (PSCs) that induce a desmoplastic reaction: by producing collagen and extracellular matrix proteins, they give rise to a dense fibrotic stroma [79,81]. Basically, this fibrotic transformation creates a “stromal fortress”, which impedes efficient drug delivery and host immune surveillance [69,80].

The rich extracellular matrix (ECM) of the TME is shaped by proteins such as collagen fibers and fibronectin, proteoglycans such as hyaluronic acid (HA), and other components, all of which form a dense fibrotic barrier which prevents therapeutics of various sizes (small molecules, immunoglobulins, and cell-based therapies) accessing the tumor [69,81].

In combination with uncontrolled angiogenesis, the desmoplastic stroma induces the formation of an abnormal network of vessels, which cause a high interstitial fluid pressure (IFP) and a poor blood supply to the tumor [69,76,79]. The resulting intra-tumoral hypoxia triggers rapidly dividing cancer cells to switch their energy generation to anaerobic glycolysis and thus produce lactic acid as a final metabolite. As a consequence, the hypoxic tumor environment turns increasingly acidic [69]. By forcing its cells to adapt their metabolism to the progressively unfavorable hypoxic and acidic environment, PC generates increasingly aggressive cancer cell clones [76].

One advantage of the abnormal intra-tumoral vasculature is its leakiness as a result of large endothelial gaps, which allow macromolecules to enter the tumor tissue. In PC, however, this enhanced permeation and retention effect (EPR), which facilitates drug delivery to tumors, is counter-balanced by the high interstitial fluid pressure [69]. Therefore, the TME’s high IFP, poor blood perfusion, and fibrotic physical barrier require novel treatment approaches including sophisticated drug delivery technologies. 

An emerging hallmark of the TME is its tumor-promoting immunosuppressive immunological milieu, which converts PC into an immunologically cold tumor and contributes to immune evasion [73,76]. It includes immune cells that reside in the TME as well as tumor-infiltrating immune cells. 

Macrophages are one cellular component of the residing immune cell fraction: since hypoxia-inducible factors trigger an M2 polarization, macrophages promote tumor growth via pro-tumorigenic signaling [79,84]. 

Despite their short lifespan, neutrophils have emerged as an important co-factor of the pro-tumorigenic TME. When tumor-associated neutrophils (TANs) are polarized by the TME towards their immunosuppressive N2 phenotype, they contribute to tumor progression, metastasis, and chemotherapy resistance via the secretion of cytokines and proteases which shape the TME [84]. For instance, matrix metalloproteinase 9 and neutrophil elastase have been shown to enhance the remodeling of the TME and the immunosuppressive milieu. Furthermore, a high neutrophil-to-lymphocyte ratio (NLR) reflecting systemic inflammation is predictive of a decreased overall survival in PDAC patients after surgical tumor excision [83].

Regulatory T cells (Tregs) contribute to the immunosuppressive TME by secreting cytokines such as tumor growth factor ß (TGF-ß) and interleukin 10 (IL-10). By inhibiting the function of cytotoxic CD8^+^ T cells, CD4^+^ T-helper cells, and NK cells, they facilitate immune evasion [84]. In fact, both the stromal barrier and immunosuppressive environment prevent cytotoxic CD8^+^ T cells (CTLs) from infiltrating the tumor and killing cancer cells via granzyme-mediated cell lysis [72]. Furthermore, the expression of programmed death ligand 1 (PD-L1) allows PC to escape recognition by CTLs and thus create an effective biological barrier for immune-mediated drug approaches [80].

A plethora of signaling molecules (cytokines such as TGF-ß, TNF-α, IFN- ß, IL-6, IL-10, IL-17, GM-CSF, CSF-3, VEGF, and miRNA-containing exosomes) mediate the cross-talk between tumor cells, immune cells, and the surrounding stroma, hence sustaining the chronic (smoldering) inflammation, and mediating the proliferation and malignant transformation of the tumor [80,81,85,86,87].

Ultimately, pancreatic cancer cells, in concert with its TME, tailor the tumor niche in such a way that it promotes tumor growth while acting as a barrier against a systemic immune response and suppresses anti-tumor immune invasion by cytotoxic CD8^+^ T lymphocytes [79].

### 3.3. Genetic Diversity and Clonal Evolution as a Biological Barrier

One of the hallmarks of pancreatic carcinoma is its genetic heterogeneity: while certain cancers such as acute myeloid leukemia (AML) are rooted in a clearly defined small range of 10–15 characteristic mutations, pancreatic tumors often harbor a large number of (on average 50–65) somatic mutations [72,88,89,90].

This broad spectrum of genetic alterations comprises both driver and passenger mutations, which affect a dozen core signaling pathways (KRAS, TP53, DNA repair, apoptosis, CDKN2A, SMAD4/TGF-ß signaling, nF-κB, PI3K/AKT/mTOR, Wnt/Notch signaling, and Hedgehog) [80,81,91,92,93]. A genomic analysis of PDAC reveals four characteristic oncogenic driver mutations that are found in a majority of patients: KRAS, TP53, CDKN2A, and SMAD4 [72,76,90,93,94,95].

This significant genetic diversity is partly based upon genomic instability as a consequence of impaired DNA repair mechanisms [94]. Genetic heterogeneity impacts several facets of PC including its tumor biology, its diagnosis, and subsequent therapy.

From a cancer biology perspective, pancreatic carcinoma is characterized by its high adaptive potential: because of its broad genetic diversity, PC exploits a large reservoir of genomic mutations and epigenetic alterations, allowing it to quickly adapt to changing environments such as hypoxia, nutrient deprivation, and acidic pH. In fact, new mutations and epigenetic changes allow cancer subclones to emerge and expand based on their mutational profile’s survival advantage in the changing TME [96,97]. Interestingly, clonal evolution is a highly complex process which undergoes continuous oscillations in the growth and atrophy of clonal lineages, which goes hand in hand with alternating clonal dominance (ADC) [96]. Ultimately, by giving rise to a highly dynamic clonal evolution, PC’s genetic diversity causes rapid tumor progression and increased metastatic potential.

PC’s complex clonal architecture does not only impact its biological development by promoting an aggressive growth, but also complicates cancer diagnosis and treatment [98]. Because of intra-tumoral differences, a single tumor biopsy is unlikely to reflect the entire tumor’s spatial genetic fingerprint and hence might be insufficient for treatment guidance and prognosis [97,98]. Furthermore, PC’s dynamic clonal evolution necessitates the continuous monitoring for early identifying new aggressive clones [98,99].

PC’s genetic heterogeneity, including the concurring expansion of genetically diverse subclones, gives rise to treatment resistance, and tumor progression or relapse. The fraction of highly malignant cancer stem cells, senescent cancer cells, and many of the cell populations in the TME contribute to the emergence of chemotherapy resistance [100]. To address this important challenge, the therapy design often incorporates a combinational therapy that targets multiple mutations to minimize the risk of escape mutations, and an adaptive therapy approach which considers the dynamics of clonal evolution [97,98]. Ultimately, inter-tumoral differences between the genetic signatures of PC urge a personalized therapy approach designed according to the genomic profile of every patient’s tumor.

### 3.4. Metastasis as a Biological Barrier

As 90% of cancer patients succumb to metastasis, targeting and eliminating metastatic cells including circulating tumor cells (CTCs) in the blood stream is an important therapeutic pillar in advanced disease [101,102]. 

Pancreatic cancer’s broad genetic diversity and its unique pro-tumorigenic TME exert a high selection pressure and, hence, drive the emergence of subclones with increased metastatic potential [96]. Certain mutations that are common in PC such as genetic alterations in the tumor-suppressor gene TP53 and loss of SMAD4 are linked to enhanced metastatic propensity [76].

Cancer metastasis provides a great example of how cancer cells evolve over time and change their phenotype by acquiring mutations that allow them to subsequentially overcome multiple biological barriers. In order to metastasize, tumor cells have to undergo a series of genetically and epigenetically driven transformations termed metastatic cascade, which provide them the ability to detach from the epithelial cell association within the primary tumor tissue, migrate through the extracellular matrix, invade the blood stream, survive within the circulation, and finally extravasate in a suitable distant organ (“homing”) to form a new tumor [103]. This cellular genomic and phenotypic transformation is called epithelial–mesenchymal transition (EMT): it allows epithelial cancer cells to lose their epithelial markers alongside their apicobasal polarity and turn into mesenchymal stem cells, which provide cancer cells the capabilities to migrate, invade blood vessels, and form secondary tumors [104,105]. EMT is often triggered by the activation of transcription factors belonging to the SNAIL, TWIST, and ZEB pathways [106,107]. The characteristics of EMT are several phenotypic changes such as the loss of epithelial markers (E-cadherin, claudins, and cytokeratin) and the expression of mesenchymal markers (N-cadherin, vimentin, and fibronectin) [101,105,106,107].

Once the activation of the EMT program enables cancer cells to detach from the primary tumor and migrate, they invade the blood stream and/or lymphatic tissues, hence turning into circulating tumor cells (CTCs). As new evidence suggests, many cancer cells can be found in the circulation even at earlier stages of disease, which underlines the necessity of targeting CTCs.

Drug delivery to (clustered) CTCs is particularly challenging: they are not only a difficult-to-catch moving target, but they also adhere to and activate platelets to form a microthrombus, which acts as a protective barrier shielding CTCs from potential systemically delivered therapeutics [102].

The last step of the metastatic cascade is the non-random homing of CTCs to specific metastatic sites. Thus far, the factors determining the timing, metastatic organotropism, and success rate for CTCs to seed and expand new metastatic sites remain elusive. Probably, a set of specific driver mutations in neoplastic cells is a prerequisite for colonization [103,108]. Furthermore, the availability of a pre-existing organ-specific favorable microenvironment, called premetastatic niche (PMN), is also required for CTCs to colonize distant tissues [101]. New evidence reveals that a primary tumor can induce the formation of PMNs by sending out tumor-cell-derived exosomes. These exosomes could carry tumorigenic micro-RNAs such as miR-200 which has been shown to enhance breast cancer cell metastasis [101,109,110].

A new hypothesis suggests a correlation between microenvironmental changes in the primary tumor and the probability of metastatic cells forming a new colony. In fact, changes in the primary tumor’s TME might induce clonal reshuffling which mobilizes those cells which are most potent to seed secondary tumors [96]. 

The therapeutic targeting of metastasis is not only challenged by the cells’ phenotypic changes in the context of EMT and the difficulties of targeting CTCs, but also by the anatomical–physiological barriers attributed to different metastatic sites.

Pancreatic cancer primarily metastasizes to the liver, and, to a lesser extent, to the lungs and bones [111]. In rare cases, it can also spread into the brain and other organs. The most common site of metastasis is the liver because of the portal vein, which carries blood from the pancreas to the liver [111]. Liver metastasis can often be treated by surgical resection, as the liver is anatomically divided into distinct easily removable segments and has an extraordinary regenerative capacity. Brain metastasis, on the contrary, is rare and difficult to treat via surgery. It has an extremely poor prognosis, and drug delivery is challenged by the blood–brain barrier [111].

From a drug delivery standpoint, metastatic sites within the bones are also difficult to access. In fact, the blood–bone barrier and the highly mineralized dense bone tissue act as a physical barrier which limit (even) drug distribution. Furthermore, the bone marrow’s (BM) dense cellular network can also impede efficient drug delivery to metastatic sites within the BM. Ultimately, the blood–bone barrier, the dense bone matrix, and the poor solubility of drugs in bone tissue make it challenging for a systemically administered drug to reach therapeutic concentrations within the bone tissue [112].

Ideally, every organ which is infiltrated by metastatic colonies requires an individual drug delivery approach, for every single metastatic site forms a different biological barrier.

In the future, a deeper understanding of the pattern of mutations and the epigenetic signature, which facilitate metastasis, might assist the design of (combinational) therapeutic strategies to prevent cancer from forming new colonies.

### 3.5. Therapeutic Strategies to Overcome Biological Barriers in Pancreatic Cancer

#### 3.5.1. Basic Therapeutic Principles

The hallmarks of pancreatic cancer—namely, its specific TME, its broad genetic diversity, and metastatic potential—explain why most therapeutic approaches have not been fruitful so far. Therefore, new therapeutic modalities are being developed to overcome these obstacles.

An effective therapeutic strategy builds upon addressing the tumor-specific challenges and optimizing the drugs’ pharmacokinetic (PK) profiles according to the drugs’ mode of administration.

In oncology, systemic (intravenous) infusion represents the main route of drug administration. For systemically applied therapeutics, bioavailability is mainly determined by renal clearance, metabolic transformation in the liver, and drug uptake into undesired organs (tissues, and compartments) and cells including the mononuclear phagocyte system (MPS). 

The MPS, which consists of macrophage-like Kupffer cells in the liver and sinusoidal macrophages in the spleen, is responsible for clearing the host system from foreign particles, with particle tagging by opsonization enhancing this process.

As a molecule’s PK profile is key for efficient drug delivery, multiple hemodynamic concepts have been developed to increase the circulation time of antibodies, peptides, and nucleic-acid-based therapeutics. Drug modifications include the attachment of polyethylene glycol, known as pegylation, and drug encapsulation into various nanocarriers.

Despite numerous clinical trials to explore the potential of new drugs, the standard therapy for pancreatic carcinoma has not changed significantly over the past two decades. 

Nevertheless, gemcitabine remains a basic pillar of the (standard) chemotherapeutic therapy [81]. It can either be infused alone, or in combination with erlotinib or the taxane nab-paclitaxel [76,113]. Paclitaxel exerts its anticancer effects by inhibiting the depolymerization of microtubules during cell division. To enhance the drug delivery properties of paclitaxel, it was reformulated and is nowadays used as nanoparticle albumin-bound paclitaxel (nab-paclitaxel) [81].

Patients with a good performance status might be eligible for the FOLFIRINOX regimen which is an effective but burdensome combination therapy based on the concurrent administration of folinic acid (leucovorin), fluorouracil (5-FU), irinotecan, and oxaliplatin [76]. In 2010/2011, FOLFIRINOX emerged as the most potent treatment for patients with advanced PC, with a median overall survival benefit of approximately 4 months compared to gemcitabine alone [113,114,115].

In 2015/2016, a new nano-liposomal formulation of irinotecan (nal-IRI, Onivyde) was approved by the FDA and EMA, and is now available for the treatment of patients with PC [116]. 

Despite the remarkable success of nab-paclitaxel, FOLFIRINOX, and liposomal irinotecan, the development of many new treatment approaches has not yielded any major breakthroughs. 

Although immunotherapy has revolutionized cancer therapy for many tumor entities, PDAC remains resistant to this modern therapy approach [79,80]. In fact, PDAC’s immune-suppressive TME, its limited immunogenicity, and the poor access of cytotoxic CD8^+^ T cells to the tumor tissue due to its stromal barrier turn PDAC into an immunologically cold tumor [74].

Novel therapeutic approaches focus on the characteristics of PC including its pro-tumorigenic immunosuppressive TME, its genetic diversity, and metastatic potential.

#### 3.5.2. Therapeutic Targeting of the TME

In pancreatic cancer, the TME typically constitutes up to 80–85% of the tumor bulk. Its fibro-inflammatory nature contributes in large part to the proliferation, aggressiveness, and chemotherapy resistance of PC [107]. Since the fibrotic stroma surrounding PC and its abnormal vasculature form a physical barrier, which slows down the perfusion of systemically administered drugs, several strategies aim to remodel the TME by softening the extracellular matrix and improving or restoring the blood supply. One approach is to use a metalloproteinase, which enzymatically degrades various components of the ECM [79]. Another option is to target the stromal ECM with pegylated human recombinant PH20 hyaluronidase (PEGPH20) to enzymatically digest hyaluronan as one of the predominant substances of the ECM and hence reprogram the dense tumor stroma [81,117]. In Phase 2 trials in 279 patients with metastatic pancreatic ductal adenocarcinoma receiving treatment of PEGPH20 plus nab-paclitaxel/gemcitabine or nab-paclitaxel/gemcitabine alone, promising results were observed (in particular, a significant improvement in the progression-free survival in patients with HA-high tumors). However, the results of a recent Phase 3 trial could not confirm any significant improvement in clinical outcomes when PEGPH20 was added to the treatment with nab-paclitaxel/gemcitabine, hence manifesting the complexity of therapeutic strategies to remodel the TME [117].

Several small leucine-rich proteoglycans (such as decorin and lumican), which are expressed by PSCs in the ECM, appear to play a role in mediating signaling across the TME. Several studies are ongoing to elaborate their precise function and their potential as drug targets [79].

Currently, various studies explore how to therapeutically target the immunosuppressive TME and convert pancreatic cancer from an immunologically cold into a hot tumor. One approach is to therapeutically regulate and steer the polarization of tumor-associated macrophages (TAMs) and neutrophils (TANs) towards their M1 and N1 phenotypes, respectively. In this context, agonistic CD40 monoclonal antibodies have emerged as a promising candidate to modulate the functional plasticity of TAMs by shifting their phenotype to the more favorable M1 polarization, which promotes the apoptosis of pancreatic cancer cells [73,76,81,118,119]. Furthermore, agonistic CD40 therapy leads to the activation of tumor-specific T cells via dendritic-cell-mediated priming, hence inducing tumor regression via multiple channels [120]. 

With the deficiency in major histocompatibility complex (MHC) I antigen presentation being one of cancer cells’ immune evasion mechanisms, an elegant new approach is to restore TME inflammation and immune cell infiltration by the systemic delivery of mRNA-encoded interleukin-2 [121].

Pancreatic stellar cells contribute to shaping the tumorigenic TME by the production of collagen and hyaluronan, hence fortifying the stromal barrier. Preclinical studies suggest that targeting PSCs by angiotensin inhibitors such as losartan inhibits the proliferation of tumor-promoting PSCs [81].

#### 3.5.3. Therapeutic Targeting of Genetic Diversity

Pancreatic cancer’s broad genetic heterogeneity and dynamic clonal evolution form a complex biological barrier, which transforms the tumor into a “continuously moving target” and inhibits an effective therapy [98,122]. 

Despite the emergence of many new therapy modalities over the past decade, cancer cell plasticity remains a major obstacle for therapeutic intervention. Importantly, a tumor’s genetic heterogeneity is not only spatial, meaning that the clonal architecture varies within the disease site, but also temporal as a result of the clonal evolution and continuous tumor remodeling [99]. Currently, the best therapeutic strategy to overcome these challenges is to apply two treatment modalities in cancer care: a combination regimen, and a personalized treatment design, which includes a targeted as well as adaptive therapy approach [99]. 

Combination therapy has already proven to be a successful strategy. In fact, FOLFIRINOX as a combination regimen allows us to simultaneously target multiple pathways, hence minimizing the risk of escape mutations and the development of treatment resistance. Unfortunately, multidrug resistance remains a major problem in cancer therapy. Cancer cell clones which overexpress efflux pumps such as permeability-glycoprotein (P-gp) have the capacity to pump out about 100 different chemotherapeutics, hence rendering therapy ineffective. Therefore, enriching chemotherapy regimens by an efflux pump inhibitor such as verapamil might help overcome multidrug resistance [123]. 

Personalized cancer treatment has emerged as a novel precision-medicine-based approach for treating every tumor individually by taking into consideration intra-tumoral variability (over time), the reciprocal influence between the neoplasm and its TME, and individual patient characteristics (including lifestyle, environmental factors, and co-morbidities) [90,98,124].

Nowadays, the availability of a broad range of monoclonal antibodies and small-molecule drugs that specifically inhibit multiple cancer signaling pathways allows us to design a personalized combination treatment based on the tumor’s genetic fingerprint. A precision medicine program called Know Your Tumor^®^ (KYT) demonstrated not only the great potential of molecular tumor profiling for tailoring a patient’s personalized therapy but also its beneficial impact on therapy outcomes: the KYT program revealed that about half of all PC patients have a mutation which is actionable and for which a targeted therapy is available. When a small patient cohort was switched from their standard treatment to a personalized targeted therapy, they demonstrated a significant improvement in their median progression-free survival [90,125].

Ideally, a personalized treatment approach includes adaptive therapy. Because adaptive therapy requires longitudinal molecular analysis to track the evolving clonal architecture, various diagnostical tools have to be implemented. These comprise next-generation sequencing (NGS), including the whole-exome sequencing (WES) of tumor samples, multiregional sequencing, single-cell sequencing, and the longitudinal analysis of liquid biopsy samples [99]. Regular genomic profiling will ensure the stringent monitoring of the tumor’s clonal evolution and allow us to adjust the targeted therapy based on the evolving mutational signature.

A revolutionary approach, which might soon become available to fight PC, is a therapeutic cancer vaccine. The innovation leaders BioNTech and Moderna alongside other biotechnology companies are working on different concepts to develop vaccines against various cancer entities.

A huge advantage of a therapeutic vaccine is its potential to simultaneously target multiple tumorigenic antigens and activate the host immune system to eradicate cancer cells. Ideally, the tumor’s mutational signature serves as template for a personalized vaccine, which addresses the patient-specific genetic heterogeneity and hence minimizes the risk of escape mutations. To be effective, a vaccine has to not only elicit a humoral response by the generation of neutralizing antibodies, but also a T-cell response, which includes the activation of CD4^+^ T-helper cells and cytotoxic CD8^+^ T lymphocytes. 

Depending on the way of identifying, selecting, and delivering tumor antigens into the patient, four vaccine technologies are being investigated: cell-based cancer vaccines including autologous vaccines based on tumor lysates, viral-based cancer vaccines using engineered viruses as the delivery vehicle for tumor antigens, peptide-based cancer vaccines, and nucleic-acid-based cancer vaccines. The latter received a lot of attention through BioNTech’s and Moderna’s focus on developing mRNA-based COVID-19 and tumor vaccines [126,127]. Their cancer vaccine approach harnesses mRNA transcripts encoding up to 20 neoantigens, which are delivered using mRNA-lipoplex nanoparticles. The neoantigen design is based on the next-generation sequencing (including WES) of tumor biopsies and subsequent in silico neoantigen prediction via bioinformatic tools [128].

Currently, several cancer vaccine candidates targeting PC are being investigated in clinical studies: VCC-001, developed by Vaccentis, is an autologous tumor vaccine harnessing a tumor lysate. It has significant potential for renal cell carcinoma, and colon cancer [129,130]. A major advantage of using a tumor lysate is that it includes all tumor antigens.

Personalized mRNA-based cancer vaccines are currently developed by BioNTech and Moderna. As exogenously administered mRNAs are recognized by endosomal and cytosolic innate immune receptors (such as toll-like receptors 7 and 8), mRNAs are rendered non-immunostimulatory by the incorporation of modified nucleosides such as pseudouridine. To enhance cellular uptake and protect mRNAs from nuclease digestion, they are embedded into lipid-based or liposomal nanocarriers [131]. 

Adjuvant autogene cevumeran (BNT122), a personalized cancer vaccine based on (uridine) mRNA-lipoplex nanoparticles, is currently enrolling patients for a randomized Phase 2 clinical trial [74]. A previously conducted Phase 1 study showed poly-specific T-cell responses and delayed tumor recurrence up to three years post tumor resection [132]. The established process of tumor sample acquisition, and subsequent cell sequencing to identify suitable neo-epitopes, finally followed by mRNA design by integrating the most immunogenic neo-epitopes, allows a rapid and scalable production of personalized tumor vaccines [133]. Two major challenges of the mRNA-based cancer vaccine approach, however, are the accurate mapping of the cancer mutanome and the selection of neoantigens on a patient-specific basis [127]. The selection is error-prone, as it depends on the quality of the neoantigen prediction algorithm and does not include all tumor antigens.

Given a rich pipeline of tumor vaccines and many recent scientific breakthroughs, it is likely that cancer vaccines will revolutionize cancer therapy within the next years.

Nowadays, the challenges arising from a neoplasm’s genetic diversity and clonal evolution can be addressed most efficiently by combining a personalized poly-therapy concept with an adaptive treatment approach.

#### 3.5.4. Therapeutic Targeting of Metastasis

The metastatic cascade comprises several consecutive steps and, therefore, offers multiple druggable targets: (a)the epithelial–mesenchymal transformation of tumor cells at the site of the primary tumor;(b)the migration of metastatic cells as circulating tumor cells (CTCs) in the blood stream and/or lymphatic tissues;(c)metastatic colonies in distant organs.

The epithelial–mesenchymal transition of cancer cells, rooted in the tumor’s cellular plasticity, allows epithelial cancer cells to acquire mesenchymal traits to enhance their migratory and invasive capabilities for forming new colonies in suitable organs [107]. With transforming growth factor β (TGF-β) being a crucial driver for EMT, inhibitors of the TGF-β pathway might reduce the metastatic potential of primary tumor cells [100,134,135]. A plethora of molecular compounds (such as losartan) are in preclinical or clinical studies, either as inhibitors of the TGF-β ligand, the TGF-β receptor, or the downstream mediators of TGF-β signaling [81,100,135].

Since Hedgehog signaling is also involved in shaping the TME and promoting metastasis, the therapeutic use of Hedgehog inhibitors is likely to slow down cancer progression [136]. Various pharmacologic Hedgehog pathway inhibitors such as vismodegib, sonidegib, saridegib, and glasdegib are being investigated in clinical trials [137].

With the mechanistic target of rapamycin (mTOR) signaling pathway enhancing the migration and metastasis of cancer cells, several mTOR inhibitors such as everolimus and temsirolimus are investigated in combinational therapy to treat PC [87,138]. Paradoxically, mTOR inhibition might favor the emergence of cancer cells that are resistant to chemotherapy. In fact, in this resistant cancer cell population, the activation of the mTOR pathway increases chemosensitivity in vitro and in vivo, hence demonstrating the variety of effects signaling pathways can elicit [139].

Tumor cells which have invaded the blood and/or lymphatic vessels are a difficult drug target. Drug delivery is dramatically complicated by the fact that CTCs are a moving target shielded by a protective layer of activated platelets. Basically, (clustered) CTCs form a microthrombus which acts as a barrier against the efficient delivery of therapeutics and recognition by the host immune surveillance system.

A recent study suggests that platelet-coated (platelet membrane biomimetic) nanoparticles bear the potential to deliver drugs to CTCs. In fact, Da and colleagues demonstrated in a cancer mouse model that the targeted co-delivery of antibodies against PD-L1 and sorafenib using platelet-functionalized nanocarriers did not only help recruit T cells to circulating CTCs, but also prevented them from forming metastasis in distant organs [102].

New findings show the “homing” of CTCs in distant organs requires a suitable microenvironment at the metastatic site to allow a new tumor to expand. 

Based on the mechanistic insights into the role of tumorigenic microRNAs and their exosomal delivery to metastatic sites where they facilitate a favorable tumor microenvironment, a promising avenue to inhibit metastasis is to deliver tumor-suppressive miRNAs that inhibit the formation of metastasis. Aside from naked miRNAs, various vehicles such as exosomes can be harnessed for the delivery of miRNAs to the primary tumor and pre-metastatic niches. In a murine prostate cancer xenograft model, atelocollagen-delivered miR-16 inhibited the formation of bone metastasis by regulating genes mediating cell-cycle control [140].

Depending on the organ, which is invaded by metastasis, specific targeted therapeutic approaches might be most promising. For instance, to specifically target bone metastasis, chemotherapeutics are conjugated with bisphosphonates, which selectively accumulate in bones due to their high affinity with hydroxyapatite [112].

#### 3.5.5. Summary of Therapeutic Approaches in Pancreatic Cancer

New mechanistic insights into the complex pathophysiology of PC have paved the way for new therapeutic approaches targeting the hallmarks of PC, which form challenging biological barriers: PC’s nourishing TME, and its genetic heterogeneity and metastatic potential (summarized in Figure 2 and Table 1). Emerging novel therapeutic strategies aim to remodel the TME by enhancing the blood supply, increasing intra-tumoral drug retention, and reversing immunosuppression. Combination and adaptive therapy will be key to addressing the challenges resulting from PC’s genetic diversity. A multitude of nanoparticle-based delivery systems are being developed to attain difficult-to-reach tumor cells including CTCs. Ultimately, the treatment of PC will become more and more tailored to the individual patient, and a set of innovative targeted/personalized medicines including immunotherapy, antibody-drug conjugates (ADCs), and cellular therapies such as chimeric antigen receptor T cells (CAR-T), and cancer vaccines holds promise in revolutionizing the treatment of this highly malignant tumor.

## 4. Chapter III: HEMOPHILIA—Biological Barriers in Rare Genetic Disorders and Gene Therapy as a Novel Therapeutic Approach in Hemophilia A and B

### 4.1. Introduction to Rare Genetic Diseases

While RNAi-based therapeutics and targeted nanomedicines turned out to be efficient for treating a broad spectrum of diseases, they are not suitable for curing genetic disorders. In contrast to cancer and infectious diseases such as HIV, hereditary (mono-) genetic disorders often require the substitution or supplementation of the missing or defective gene by an imported (functional) transgene. For this purpose, a curative therapeutic does not only have to pass the plasma membrane of a target cell but also overcome the nuclear envelope to gain access to the cellular genome. In this context, gene therapy in its most fundamental form introduces genetic material into target cells via viral and non-viral vectors [141].

Nature has invested considerable energy and effort to protect cells against viral entry [142]. Paradoxically, the transfer of functional genes into cells can be facilitated by viruses re-engineered to perform as vectors. Currently, in clinical trials, viral vectors are being used to transfer genes to treat cardiovascular, muscular, metabolic, neurological, hematological, and ophthalmological diseases, as well as infectious disorders and cancers [143]. The most efficient viral vectors to emerge from preclinical and clinical studies are adenovirus vectors, adeno-associated virus (AAV) vectors, retroviral vectors, and lentiviral vectors (as a subtype of retrovirus) [144]. 

### 4.2. Biology of AAV-Mediated Gene Therapy as Novel Therapeutic Approach

Viral-vector-mediated gene therapy has coalesced around two primary approaches, in vivo AAV gene transfer (episomal gene transfer) or ex vivo Lenti virus (gene insertion). This section will focus on AAV gene transfer.

AAV belongs to the genus Dependoparvovirus within the family Parvoviridae [145,146]. AAV was first described in the mid-1960s [147]. After its discovery, subsequent research studies provided the foundational knowledge that led to the use of AAV as a gene delivery vector. Understanding genome composition, infectious latency, and virion assembly allowed for the cloning of the wild-type AAV2 sequence into plasmids, which enabled genetic studies [146]. Importantly, wild-type AAV does not cause any human diseases [148,149]. Thus, as a replication-incompetent vector, with a predictable tropism for selected tissues and the capability to be endocytosed and unpacked within the target cell nucleus, AAV has become a workhorse vehicle for in vivo gene transfer therapy [146,150,151]. 

The AAV vector has distinctive features that are beneficial in clinical applications, including broad (cell) tropism, low immunogenicity, and ease of production. AAV is also non-pathogenic, rarely integrates into the host chromosome, and results in the long-term expression of the transgene [141,152]. 

As a parvovirus, AAV carries a single-stranded DNA genome of 4.7 kb, encoding two genes (rep and cap) which are flanked by two inverted terminal repeats [145,146,149]. While the rep gene produces proteins required for viral replication, the cap gene encodes for the capsid proteins VP1, VP2, and VP3 [141,146].

Studies have shown that AAV infects target cells by binding primary receptors and co-receptors on the cell surface, which triggers their endocytosis into endosomes [153]. The biology of viral entry into the target cells, transport across the nuclear membrane, and uncoating and release of the transgene is complex and incompletely understood [154].

At least nine discrete steps have been identified as important in the AAV vector transduction pathway [141]. The first step involves the adeno-associated virus (AAV) vector virions binding to receptors and co-receptors on the surface of target cells. The AAV vectors are endocytosed within the cells (step 2). Following the release from endosomes, AAV virions are either ubiquitylated and targeted for proteasome-mediated degradation (step 3) or intracellularly trafficked to the nucleus (step 4). Once in the nucleus, AAV virions are uncoated and the AAV transgene is released (step 5). The AAV single-stranded DNA genome is then converted into double-stranded DNA (step 6), followed by transcription (step 7) and the nuclear export of mRNA (step 8) for the translation and expression of the therapeutic transgene (step 9) [141].

How AAV virions enter cells to efficiently deliver their gene cargo remains poorly understood [155]. Tissue and cell tropism is determined by the capsid structure, which is composed of three proteins (VP1, VP2, and VP3) [141,149]. Capsid variants differentially engage cellular entry factors. For several AAV serotypes, viral entry is initiated via capsid-specific sugar moieties on the cell surface [156]. Many AAV serotypes bind to specific receptors on the surface of target cells, with several of the serotypes reported to bind to a primary cell surface receptor, followed by interaction with a secondary receptor that appears to enable viral entry. Primary cell surface attachment receptors identified include heparan sulfate proteoglycans for AAV serotypes 2, 3, and 6; N-terminal galactose for AAV9; and specific N- or O-linked sialic acid moieties for AAV1, 4, 5, and 6 [157]. Secondary receptors include fibroblast growth factor receptor (FGFR) and possibly integrin for AAV2; hepatocyte growth factor receptor (c-Met) for AAV2 and 3, and platelet-derived growth factor for AAV5, which is also modified by sialic acid [157]. To date, no single receptor common to all AAV serotypes has been identified [157]. 

### 4.3. Short History of Gene Therapy

The concept of gene therapy has been hypothesized for decades. Gene transfer has received determined interest with the introduction of recombinant DNA technology and the ability to transfer and express exogenous genes in mammalian cells [144]. The first clinical trials were carried out in the late 1980s. At that time, gene therapy was predicted to become a treatment for monogenic diseases in less than a decade [157]. However, during the next two decades, several obstacles and fatal outcomes in patients have moderated the excitement for gene therapy.

In 1999, an 18-year-old who received gene therapy to treat a metabolic disorder died as a result of a lethal immune reaction to the viral vector. Just a few years later, in five children suffering from a rare condition named SCID-X1, gene therapy turned out to activate an oncogene and hence cause insertional mutagenesis, leading to leukemia. Fifteen years later, a sixth child developed leukemia, demonstrating the long-term risk of oncogenesis [143,158,159,160]. 

These fatal incidences basically brought the development of gene-therapy-based therapeutics to a halt until a renaissance emerged a few years ago. Finally, after several decades of intense research, the first EMA-approved gene therapy became available in 2012 to treat adults with a hereditary metabolic condition known as familial lipoprotein lipase deficiency (LPLD): based on a recombinant adeno-associated virus (rAAV) vector, alipogene tiparvovec (Glybera) transduces muscle cells with a functional gene of LPLD [146,160,161].

Today, rAAVs have advanced to the world’s leading platform for gene therapy products [146].

### 4.4. Biological Barriers for the Application of Gene Therapy

Despite these recent breakthroughs, five important barriers continue to constrain progress (visualized in Figure 3 and described in Table 2): 

**Transgene Size**: The first restriction to the application of gene therapy is the size of the transgene, which needs to fit into the delivery vector. For recombinant AAV vectors, the DNA packaging size is limited to about 4.7 kb, hence requiring transgenes to be engineered accordingly [161,162]. If the vector genome exceeds the vector capacity, the transgene expression and protein folding might be adversely affected [152].

**Vector Organotropism and Uptake**: Second, vector delivery to the target cells, and vector uptake, intracellular transport, and uncoating remain a problem. Vector organotropism and uptake by the target tissue are prerequisites for the delivery of sufficient genetic material to allow for therapeutic levels of transgene expression [146]. Organotropism and cellular targeting can be enhanced by engineering cell-targeting viral capsids and gene promoters.

**Vector Genome Integration and Persistence**: Third, vector genome integration and persistence are a challenge. With AAV vectors being non-integrative, the active transgene exists as a double-stranded DNA episome, which will be lost during cell division [154,163]. Additionally, after the uncoating process, the transgene expression may be further regulated by the epigenetic state of the vector capsid [152,164].

**Sustainable Gene Expression**: Fourth, the transcriptional expression of the transgene needs to be sustainable [163]. As the expression of the transgene following rAAV-mediated delivery is driven by episomal genomes, transgene levels often decline over time due to the loss of episomes during cell division [152]. Additionally, transgene expression, whether driven from integrated or episomal vector genomes, can be silenced or terminated by epigenetic modifications to the vector genome [152,165]. As a result, transgene expression might fade over time, necessitating redosing. However, redosing remains a major barrier to the success of gene therapy because of the host’s immune response [142].

**Immune System**: Last but not least, an important caveat of applying AAV-vector-based gene therapy is the host’s innate and adaptive (both humoral and cellular) immune response against the AAV capsid antigens and the transgene product.

Three immunological barriers to gene therapies need to be considered: immunogenicity toward a. the vector delivery system (AAV serotype), b. the transduced target cells, and c. the immunogenicity specific to the transgene product [142,145,161,163]. 

As the AAV vector is engineered based on a wild-type parvovirus which naturally infects humans, it often elicits a B-cell response, resulting in neutralizing antibodies [145,166]. Immune responses in the GENEr8-1 trial (NCT03370913) for hemophilia A were predominantly directed toward the AAV5 capsid, and all study subjects developed a durable response against the AAV5 serotype used in the study. This high-titer antibody response will likely prevent redosing with the same serotype or other AAV serotypes due to the high degree of homology between serotypes.

AAV vectors can not only induce an adaptive immune response, but also activate the innate immune system. For instance, pathogen recognition receptors (such as toll-like receptors), which are located on the surface or in the cytosol of host cells, or within endosomal compartments, recognize pathogen-associated molecular patterns and induce an inflammatory interferon response [145,161]. 

Since vector immunogenicity has dose-dependent characteristics, the application of lower vector doses might reduce unfavorable humoral immune responses [142,145,161]. Furthermore, as practiced in current trials, concomitant immunosuppressive therapy with prednisolone improves the outcomes of AAV-mediated gene transfer [145].

b.Transduced cells can elicit capsid T-cell responses in humans, which has been implicated in effecting the duration of the transgene expression [142]. When a T-cell response leads to the rejection and apoptosis of transduced cells, the transgene expression is abrogated, which renders gene therapy ineffective [145].c.The transgene itself can also induce an immune response [161,163]. In the worst case, antibodies do not only neutralize the soluble protein, which is produced as a transgene product, but also interfere with substitution therapies that deliver the missing protein [145].

These immunological barriers should be considered additive with cellular uptake barriers. The complexity of AAV vector gene transfer reflects the rationale for the vector design meeting natural cellular defenses and mechanisms of homeostasis.

**Table 2 pharmaceutics-16-01207-t002:** This provides an overview of all biological barriers which are hurdles for effective rAAV-mediated drug delivery.

rAAV-Mediated Gene Therapy: Challenges for Drug Delivery		References
**Transgene Size**	Vector packaging size limited to 4.7 kb Transgene engineering of FVIII cDNA	[162,163]
**Vector Uptake and Organotropism**	Vector uptake is organ-/cell-dependent Capsid modifications aiming at retargeting AAV tropism permit tissue-specific gene transfer Selection of tissue-/cell-specific promoters can direct transgene expression to cells of interest	[142,150,162,164,167]
**Vector Genome Integration and Persistence**	rAAV delivers transgenes as non-integrative episomes, which are lost upon cell division Lowering vector doses reduces the risk of insertional mutagenesis	[155,164,165]
**Gene Expression Sustainability**	Transgene expression, driven by circular episomes, depends on episomal persistence, epigenetic modulation and choice of promoters Transgene expression is challenged by adaptive immune responses to the transduced cells and/or transgene itself	[143,163,164,166,167]
**Immunological Barrier**		
	Humoral antibody-mediated immune response to viral vector	[146,162,164,167]
	Cellular CTL-mediated immune response to transduced host cells	[143,146,162,167]
	Humoral and cellular immune response to transgene	[146,162,164,167]

### 4.5. Therapeutic Application of Gene Therapy in Hemophilia A and B

Recently, several important technical barriers have been overcome, paving the way for more than thirty FDA-approved gene therapies which are available now to treat various diseases. 

In the past years, gene therapy was approved for the treatment of several rare diseases and introduced a curative approach for hemophilia A and B [145,162,166,168,169,170].

Hemophilia is a hereditary recessive X-linked coagulopathy resulting from decreased or absent functions of the genes that express human clotting factor VIII or IX (FVIII/FIX). A deficiency in factor VIII leads to hemophilia A, whereas a lack of factor IX manifests as hemophilia B. While FIX is endogenously produced in hepatocytes, FVIII is predominantly secreted from sinusoidal endothelial cells in the liver [161,162,171].

The bleeding in hemophilia is characterized by a preponderance of bleeding in joints and muscle. Repetitive joint bleeds lead to hemophilic joint arthropathy, which causes significant morbidity [161,162,171]. Bleeding into closed spaces such as intracranial bleeds can be fatal [148,152]. The bleeding phenotype severity is predicted by factor activity. Patients with severe disease have <1% of normal factor activity and experience frequent spontaneous bleeds [162].

The current standard of care for patients with severe hemophilia A and B requires lifelong prophylaxis with an exogenous replacement factor, which is available both as plasma-purified and recombinant protein (FVIII for hemophilia, and FIX for hemophilia B) [152,168]. Although regular intravenous injections with a factor concentrate raise the plasma levels of the clotting factor, it is challenging to attain and maintain physiologic levels of factor VIII/IX. The development of extended half-life products improved the bioavailability of intravenously injected plasma factor and reduced the dosing frequency, but (lifelong) therapy remained burdensome [162].

Recently, the approval of emicizumab (Hemlibra), a bispecific monoclonal antibody bridging activated factor IX and factor X, and hence mimicking factor VIII, has changed the treatment landscape of hemophilia A by allowing (less painful) subcutaneous administration [162,171].

Despite the emergence of emicizumab, substitution therapy remains burdensome: as patients require (lifelong) regular intravenous or subcutaneous injections, compliance is still challenging. Furthermore, all these treatments are extremely expensive and can easily cost millions of dollars for a patient’s lifetime treatment [171].

Based on these unmet needs, a curative approach mediated by gene therapy is desirable. Recent evidence has demonstrated that raising factor levels above 10–15% reduces the bleeding phenotype substantially, hence providing the rationale for gene therapy [172]. 

In the past 2 years, the market authorizations of gene therapy for hemophilia A and B revolutionized the treatment of hemophilic patients by offering them a curative approach, hence eliminating the need for regular substitution therapy [145].

In 2022, the FDA approved etranacogene dezaparvovec (Hemgenix) for the (one-time) treatment of adult patients with hemophilia B, which marked a milestone in hemophilia therapy [162,169]. Delivered by an AAV5 vector, the Hemgenix transgene encodes the highly effective Padua variant (R338L) of human factor IX and targets hepatocytes.

In 2023, valoctocogene roxaparvovec (Roctavian) received market authorization from the FDA for the treatment of adults with severe hemophilia A [162]. In hemophilia A, the applicability of gene therapy was hampered by the size of coagulation FVIII: as its full-length cDNA amounts to 7 kb and, hence, exceeds the AAV vector capacity, the transgene needed to be adjusted. Therefore, the AAV5 vector of Roctavian, which—equipped with a hybrid liver-selective promoter—targets hepatocytes, carries a B-domain deleted version of the human coagulation FVIII gene [152,161,162]. Amounting to 4.9 kb, the vector size slightly surpasses the vector packaging capacity, which might impact the transgene expression and protein folding [152].

The hemophilia gene therapy clinical trial data and follow-up observations provide valuable insights into the long-term safety and efficacy of Hemgenix and Roctavian.

rAAV vectors were selected due to their non-integrative biology. The episomal form of the transgene dramatically reduces the risk of genotoxicity from insertional mutagenesis. However, as long-term animal data suggest, integrative events can still occur, with the risk of insertional mutagenesis correlating with the applied vector dose [152,161]. Therefore, reducing the vector dose and providing concomitant immunosuppressive therapy help mitigate the risk of genotoxicity and adverse immune reactions. In fact, long-term safety continues to accrue in both hemophilia A and B gene therapy trials with few serious safety events reported after the first year post vector exposure [173].

In terms of long-term effectiveness, Hemgenix and Roctavian differ. While Hemgenix provides sustainable transgene expression of FIX, Roctavian faces challenges with declining plasma levels of FVIII [162]. The HOPE-B study (NCT03569891), in which 54 patients with moderate to severe hemophilia B were treated with Hemgenix, demonstrates stable transgene expression at 1.5 years, with mean plasma FIX activity levels of 36.9% nearing the lower range of normal levels. In fact, hemophilia B gene therapy benefits from the fact that the target hepatocytes natively produce FIX protein. Furthermore, the used FIX-Padua variant transgene yields an about eight-fold higher specific activity compared to wild-type FIX. In addition, the relatively low applied vector dose of Hemgenix did not induce any unfavorable T-cell response, which might have lowered transgene expression [145].

Although having demonstrated significant efficacy in increasing FVIII plasma levels and reducing bleeding episodes, the effectiveness of Roctavian might not be sustainable. In fact, a 2-year longitudinal analysis of Roctavian showed that the trajectory of FVIII activity exhibited first-order elimination kinetics starting at week 76. The estimated typical half-life of the Roctavian transgene-derived FVIII production (as calculated from modeled data) was 123 weeks, demonstrating that the transgene expression is fading over time and might necessitate redosing [174]. Thus far, however, redosing is not possible due to neutralizing antibodies which would eliminate the vector upon the application of a second dose.

The differences in effectiveness between Roctavian and Hemgenix might be rooted in the nature of the transduced target cells. As FVIII is not endogenously produced by hepatocytes (but sinusoidal endothelial cells), the transduced hepatocytes might not have the capacity to produce the complex FVIII protein and therefore act as a biological barrier. Moreover, in sinusoidal endothelial cells, FVIII is co-expressed with its chaperone protein von Willebrand factor (VWF) which does not occur during hepatocyte expression. Taken together, FVIII expression may behave differently from FIX due to the size differences between the proteins and the choice of a non-native target cell.

Viral-vector-based gene therapy proved to be effective for overcoming both the cell plasma membrane and nuclear envelope as important biological barriers to gain access to the cellular genome. The engineering of viral vectors has rendered gene therapy safe and effective based on the targeted cellular delivery of transgenes.

After the failure of gene therapy 1.0 in the 1990s, the engineering of more powerful and safe viral vector platforms paved the way for the success of gene therapy 2.0 in the last decade. Compared to gene therapy 1.0, the currently used AAV vectors are more effective due to the targeted cellular delivery of transgenes and the optimization of transgenes for vector packaging and enhanced gene expression [175]. Since AAV induces episome-driven transgene expression, the safety profile nowadays is favorable with a low risk of insertional mutagenesis [152].

Both Roctavian and Hemgenix emerged as formidable examples of how gene therapy 2.0 transformed a transfusion-dependent burdensome condition into a functional cure [145].

Currently, new approaches which optimize or even substitute the potentially immunogenic viral vector as the delivery vehicle are being explored in various studies: Engineering cell-type-specific promoters and vector capsids might permit us to restrict the transgene expression to the cells of interest [163]. Recently, extracellular vesicles (EVs) have been emerging as a new delivery vehicle for transgenes. Being non-immunogenic, inert, biodegradable, available from all cells, and equipped with the capability to pass diverse biological barriers (including the blood–brain barrier), EVs have the potential to evolve into the most promising delivery vehicle for gene therapy [167].

Today, gene therapy 3.0 is on the verge: revolutionary gene-editing-based technologies such as clustered regularly interspaced short palindromic repeats (CRISPR/Cas9), zinc-finger nucleases (ZFNs), and transcription activator-like effector nucleases (TALEN) will most likely form the pillar of the next generation of gene therapy [176,177,178,179].

With about 500 gene therapy clinical trials ongoing in 2024, many new therapies will probably soon become available to treat and cure hereditary genetic diseases—with an estimated annual FDA approval rate of 10–20 gene therapies starting in 2025 [143,180,181].

## 5. Concluding Remarks

This review offered insights into three different disease areas, which are globally highly relevant from a clinical, scientific, and socioeconomic perspective. Studying these different disease areas allowed us to identify and describe the overarching principles of biological barriers. Moreover, as research in HIV, cancer, and rare genetic diseases has given rise to many innovative technological advancements and breakthroughs, these disease entities are particularly suitable for highlighting novel drug delivery approaches.

The major insights we obtained include the following:The relevance of the host immune system as a predominant biological barrier;The importance of a targeted, personalized, and combinational therapy approach;The emergence of new druggable targets in HIV, cancer, and rare diseases;The development of new cutting-edge therapy modalities (such as RNAi-based therapeutics, mRNA-based cancer vaccines, and AAV-based gene therapy) in all three disease areas we visited today.

In the context of biological barriers, the host immune system emerged as a significant biological barrier, which plays a predominant role in the treatment of infectious diseases, cancer, and genetic diseases. 

For a treatment to be effective and cause as few undesired adverse effects as possible, a patient-tailored targeted therapy is key. 

Moreover, a combination therapy approach is necessary in order to overcome immune evasion mechanisms and attain more sustainable therapeutic effects: while combination antiretroviral therapy (cART) transformed HIV into a chronic condition, FOLFIRINOX as a combination treatment proved to be one of the most promising therapies for PC. And even gene therapy is only effective in combination with concomitant prednisolone application. 

Over the past decade, the more profound understanding of the disease pathophysiology of AIDS and cancer allowed us to identify a broad range of new druggable targets. These include vaginal epithelial cells (for HIV), and CAFs, PSCs, TAMs, and TANs as key players of the TME (for pancreatic cancer).

Technological advancements have paved the way for many new exciting therapy modalities which broaden the physicians’ arsenal of weapons to fight disease. Recently approved or emerging technologies include RNAi-based therapeutics, AAV-based gene therapy, gene-editing-based gene therapy approaches, exosomal delivery strategies, and personalized cancer vaccines harnessing mRNA technology.

Although the emergence of innovative therapeutic approaches empowers us to overcome a broad spectrum of biological barriers, various obstacles remain. These non-biological barriers include the implementation of novel therapies into standardized clinical practice as well as their incorporation into clinical guidelines. Furthermore, (market) access and socioeconomic strategies including reimbursement are required to not only cover the expensive therapies but also make innovation available for patients all over the world.

## Figures and Tables

**Figure 1 pharmaceutics-16-01207-f001:**
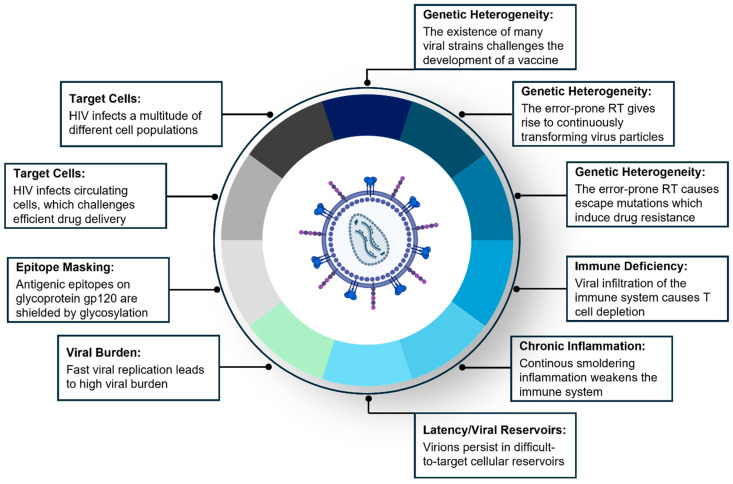
Biological barriers forming hurdles for HIV treatment. The success of HIV in establishing a worldwide pandemic is largely attributed to the multitude of biological barriers that HIV harnesses to evade both the host immune system and different treatment approaches. Its genetic heterogeneity, its deep infiltration and paralysis of the host immune system, the viral infection of various cell populations, and viral latency as provirus are major factors that explain why the development of an effective vaccine to eradicate the pandemic remains elusive.

**Figure 2 pharmaceutics-16-01207-f002:**
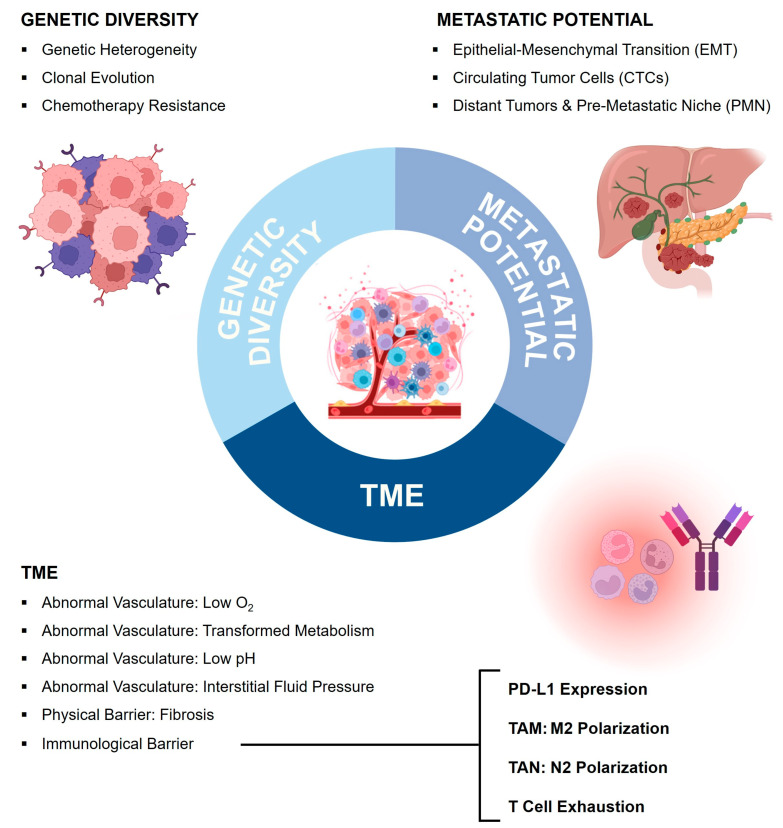
Hallmarks of pancreatic cancer and overview of biological barriers which challenge treatment success. The tumor microenvironment, genetic diversity, and metastatic potential are hallmarks of pancreatic cancer which contribute to the malignancy and treatment resistance of pancreatic cancer. At the same time, by forming a variety of biological barriers, they impede effective treatment.

**Figure 3 pharmaceutics-16-01207-f003:**
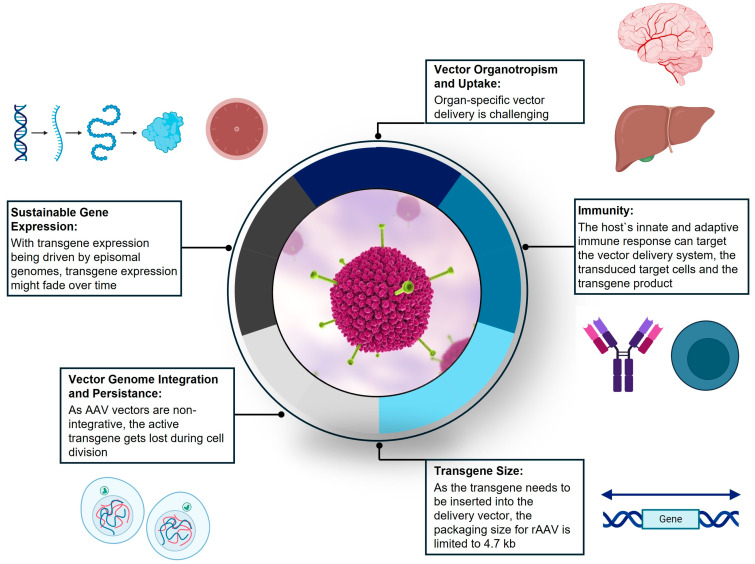
Biological barriers which challenge AAV-mediated gene therapy. The host immune system, the vector organotropism, and the sustainability of transgene expression are challenging barriers for the successful application of AAV-based gene therapy. Careful transgene and vector design contribute to overcoming these barriers, hence paving the way for the success of gene therapy.

**Table 1 pharmaceutics-16-01207-t001:** This table provides an overview of the hallmarks of pancreatic cancer and the biological barriers which challenge the development of effective treatments.

Hallmarks of Pancreatic Cancer	Challenges for Drug Delivery		References
**Tumor Microenvironment**	**Abnormal Vasculature:** **Poor Perfusion & Low Oxygen**	Poor access to target cells	[76,79,109]
	**Abnormal Vasculature:** **IFP**	Pressure gradient	[79,109]
	**Abnormal Vasculature:** **Acidic pH**	Degradation	[98]
	**Fibrotic Stroma**	Physical barrier	[79,109]
	**Immunologic Barrier**	Immunosuppressive milieu, PD-L1 expression	[76,79,80,81,83]
**Genetic Diversity and Clonal Evolution**	**Tumor Heterogeneity**	Multitude of targets Differences between patients	[93,94,97,98,99]
	**Clonal Evolution**	Changing targets and escape mutations	[97,98]
	**Chemotherapy Resistance**	Efflux Pumps, mTOR inhibition	[98,99,100,109,140]
**Metastatic Potential**	**EMT**	Changing targets due to phenotypic alterations	[109]
	**Circulating Tumor Cells**	Moving targets & protection by a shield of platelets	[102]
	**Distant Organs** ▪ **Brain:** **Blood-Brain Barrier** ▪ **Bone:** **Blood-Bone Barrier,** **poor bone vascularization,** **poor access to bone marrow**	Physical barriers (BBB) Uneven poor drug distribution No access to target cells	[110,113]

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
