# Peer review of "Biological Barriers for Drug Delivery and Development of Innovative Therapeutic Approaches in HIV, Pancreatic Cancer, and Hemophilia A/B"

_pharmaceutics, 2024, doi:10.3390/pharmaceutics16091207_

Round 1
Reviewer 1 Report
Comments and Suggestions for Authors
I found this paper interesting to read, and a comprehensive review that introduces development of innovative therapeutic approaches to overcome biological barriers in infectious diseases, cancer, and genetic disorders. I believe the paper should be published in this journal. However, at the end I would like to make some comments on these ideas, that may be helpful for improving the manuscript.
1. It is not clear why the three diseases HIV, pancreatic cancer and hemophilia A/B are selected here. Any logical explanation? It seems like three different topics. No logical connection.
2. The connection of the so-called biological barriers to the following therapeutic strategies is very weak. Please strengthen this key point.
3. "The broad range of biological barriers can be divided into macroscopic and cellular obstacles." what do you mean macroscopic obstacles? Barriers can be divided into blood, tissue, cell....
4. More specific examples should be included as figures. Current figures are too general.
5. More advantages of drug delivery systems should be introduced in an independent section, such as circulation, tissue targeting, cellular uptake (https://doi.org/10.1016/j.addr.2023.114895). This is closely related to the main topic of overcoming biological barriers.
6. The authors introduced the PEGPH20. The authors should update the information. PEGPH20, a pegylated recombinant human hyaluronidase, degrades hyaluronan (HA) and reprograms the dense tumor stroma. In Phase II trials in 279 patients with metastatic pancreatic ductal adenocarcinoma receiving treatment of PEGPH20 plus nab-paclitaxel/gemcitabine or nab-paclitaxel/gemcitabine, the promising results were observed, particularly big improvement in PFS in patients with HA-high tumors. But the recent Phase III results concluded no improvement of clinical outcomes of nab-paclitaxel/gemcitabine by adding PEGPH20. This suggest that the complexity of remodeling TME strategies (https://ascopubs.org/doi/abs/10.1200/JCO.2020.38.4_suppl.638).
Author Response
Dear Reviewer,
we would like to thank you very much for your feedback. We addressed your comments in the enclosed rebuttal letter.
We are grateful for your comments as they definitely helped improve our manuscript!
Best wishes,
Emre

Reviewer 2 Report
Comments and Suggestions for Authors
In this review, the authors describe several biological barriers in important diseases, such as HIV, pancreatic cancer, and provided insights into new therapeutic technologies. The paper is generally well written, and the data presented are of great important to the basic and applied fields of drug development. The scientific writing is very good and easy to follow.
A point of concern is why the paper does not include illustration (graphics) on “Therapeutic Strategies (Page 13, section 2.5)”? Such graphics especially of drug-delivery approaches for pancreatic cancer could improve the presentation of the work. Although adding such artwork is required the paper can be accepted on its current form.
Another points are:
- The reference 71. It looks like something missing. At least add URL of the agency.
- The references 105 and 106 can be omitted where the reference 104 is sufficient.
Author Response
Dear Reviewer,
we would like to thank you very much for your feedback. We addressed your comments in the enclosed rebuttal letter (all our answers are highlighted in blue).
We are grateful for your comments as they definitely helped improve our manuscript!
Best wishes,
Emre

Reviewer 3 Report
Comments and Suggestions for Authors
The article titled “Development of Innovative Therapeutic Approaches to Overcome Biological Barriers in Infectious Diseases, Cancer, and Genetic Disorders” has been carefully reviewed. It focuses on exploring innovative therapeutic approaches to overcome biological barriers in three different disease entities. These entities include human immunodeficiency virus (HIV), which is a predominant infectious disease, pancreatic carcinoma, one of the most lethal solid cancers, and hemophilia A/B, as hereditary genetic disorders. The article aims to showcase promising therapeutic approaches designed to cross disease-specific biological barriers and effectively treat these diseases. The authors highlight the significance of selecting the right drug category and delivery vehicle, mode of administration, and therapeutic targets to overcome various biological barriers and prevent, treat, and cure disease. However, the article would benefit from some improvements.
1. The current text contains several areas for improvement. Firstly, the section organization is unclear, and there is inconsistency in the terminology used, with references to "subsection," "chapter," and "introduction section." This could lead to confusion for readers and should be addressed for better coherence.
2. Additionally, the figures included in the article need to be more captivating and engaging to maintain the readers' interest. Consider the inclusion of schematic figures that are visually appealing and encourage readers to delve deeper into the content.
3. Furthermore, the number of references provided is insufficient, and it is recommended that recent studies be included to enhance and expand upon the points made in the article.
4. Regarding the discussion of diseases within the topic, it is advised to allocate separate sections for each disease to avoid confusion. Presenting each disease as an individual topic and providing expanded discussion would facilitate a clearer understanding for the readers.
5. Finally, it is essential to consider how readers can derive valuable information from the figures independently of the accompanying text. Efforts should be made to sustain readers' interest throughout the entirety of the figures, not just the initial portion.
Comments on the Quality of English LanguageModerate editing of English language required.
Author Response

(The authors gave the same response as above.)

Round 2
Reviewer 1 Report
Comments and Suggestions for Authors
The authors addressed the concerns well.
Reviewer 3 Report
Comments and Suggestions for Authors
The authors made significant revisions based on the reviewers comments, the MS can accept for publication.
Comments on the Quality of English Language
Minor editing of English language required.